# A Weakly Supervised Deep Learning Method for Guiding Ovarian Cancer Treatment and Identifying an Effective Biomarker

**DOI:** 10.3390/cancers14071651

**Published:** 2022-03-24

**Authors:** Ching-Wei Wang, Yu-Ching Lee, Cheng-Chang Chang, Yi-Jia Lin, Yi-An Liou, Po-Chao Hsu, Chun-Chieh Chang, Aung-Kyaw-Oo Sai, Chih-Hung Wang, Tai-Kuang Chao

**Affiliations:** 1Graduate Institute of Biomedical Engineering, National Taiwan University of Science and Technology, Taipei 106335, Taiwan; cweiwang@mail.ntust.edu.tw (C.-W.W.); m10823101@gapps.ntust.edu.tw (Y.-A.L.); m11023111@mail.ntust.edu.tw (C.-C.C.); m11023001@mail.ntust.edu.tw (A.-K.-O.S.); 2Graduate Institute of Applied Science and Technology, National Taiwan University of Science and Technology, Taipei 106335, Taiwan; d10522201@mail.ntust.edu.tw; 3Department of Gynecology and Obstetrics, Tri-Service General Hospital, Taipei 11490, Taiwan; obsgynchang@gmail.com (C.-C.C.); raynhsu@gmail.com (P.-C.H.); 4Graduate Institute of Medical Sciences, National Defense Medical Center, Taipei 11490, Taiwan; 5Department of Pathology, Tri-Service General Hospital, Taipei 11490, Taiwan; b93401052@ntu.edu.tw; 6Institute of Pathology and Parasitology, National Defense Medical Center, Taipei 11490, Taiwan; 7Department of Otolaryngology-Head and Neck Surgery, Tri-Service General Hospital, Taipei 11490, Taiwan; chw@ms3.hinet.net; 8Department of Otolaryngology-Head and Neck Surgery, National Defense Medical Center, Taipei 11490, Taiwan

**Keywords:** weakly supervised learning, ovarian cancer, precision oncology, deep learning

## Abstract

**Simple Summary:**

Molecular target therapy, i.e., antiangiogenesis with bevacizumab, was found to be effective in some patients of epithelial ovarian cancer. Considering the cost, potential adverse effects, including hypertension, proteinuria, bleeding, thromboembolic events, poor wound healing and gastrointestinal perforation, and no confirmed and accessible biomarkers for routine clinical use to direct patient selection for bevacizumab treatment, the identification of new predictive methods remains an urgent unmet medical need. This study identifies an effective biomarker and presents an automatic weakly supervised deep learning framework for patient selection and guiding ovarian cancer treatment.

**Abstract:**

Ovarian cancer is a common malignant gynecological disease. Molecular target therapy, i.e., antiangiogenesis with bevacizumab, was found to be effective in some patients of epithelial ovarian cancer (EOC). Although careful patient selection is essential, there are currently no biomarkers available for routine therapeutic usage. To the authors’ best knowledge, this is the first automated precision oncology framework to effectively identify and select EOC and peritoneal serous papillary carcinoma (PSPC) patients with positive therapeutic effect. From March 2013 to January 2021, we have a database, containing four kinds of immunohistochemical tissue samples, including AIM2, c3, C5 and NLRP3, from patients diagnosed with EOC and PSPC and treated with bevacizumab in a hospital-based retrospective study. We developed a hybrid deep learning framework and weakly supervised deep learning models for each potential biomarker, and the experimental results show that the proposed model in combination with AIM2 achieves high accuracy 0.92, recall 0.97, F-measure 0.93 and AUC 0.97 for the first experiment (66% training and 34%testing) and high accuracy 0.86 ± 0.07, precision 0.9 ± 0.07, recall 0.85 ± 0.06, F-measure 0.87 ± 0.06 and AUC 0.91 ± 0.05 for the second experiment using five-fold cross validation, respectively. Both Kaplan-Meier PFS analysis and Cox proportional hazards model analysis further confirmed that the proposed AIM2-DL model is able to distinguish patients gaining positive therapeutic effects with low cancer recurrence from patients with disease progression after treatment (*p* < 0.005).

## 1. Introduction

Globally, ovarian cancer is the most common cancer-related cause of death from gynecological tumors in women [1] but lacks methods recommended for screening and early diagnostics of this disease [2]. Around 90% of primary ovarian cancers are of epithelial origin. Epithelial ovarian cancer (EOC) is classified into serous, mucinous, endometrioid, clear cell, transitional cell, mixed epithelial, undifferentiated and unclassified subtype [3]. Peritoneal serous papillary carcinoma (PSPC), though managed according to EOC therapeutic principles, has been variably considered as an EOC counterpart [4]. The current standardized treatment for EOC is optimal cytoreductive surgery plus platinum-based chemotherapy. However, with the development of chemotherapy-resistant and refractory diseases, the sensitivity of chemotherapy has decreased [5]. There are many new drugs under development, and they are undergoing clinical trials aimed to evaluate their efficacy in the treatment of EOC, such as antiangiogenesis, inhibitors of growth factor signaling, poly-ADP-ribose polymerase inhibitors (PARP) inhibitors, or folate receptor inhibitors [2].

Tumor cells and tumor stroma both produce a variety of proangiogenic factors, such as vascular endothelial growth factor (VEGF), bFGF, interleukin 8 (IL-8), G-CSF and GM-CSF, which are designed to promote angiogenesis [6]. Angiogenesis promotes several pathophysiological conditions which are important for tumor cell growth, metastasis and also associated with chronic inflammation [7]. The key factor in the development of a tumor pathological vascular network is VEGF and its signal transduction pathway [8]. VEGF is also related to the formation of ascites in patients with EOC. Therefore, inhibiting pathological angiogenesis has become one of the new treatment options that has been widely tested in the treatment of EOC with promising therapeutic effects [2]. Bevacizumab is an anti-VEGF antibody, and its use in the first and second-line therapy of EOC is well established [9].

Considering the cost, potential side effects such as hypertension, proteinuria, bleeding, thromboembolic events, poor wound healing, and gastrointestinal perforation [10] and no confirmed and accessible biomarkers for routine clinical use to direct patient selection for bevacizumab treatment, the identification of new predictive method remains an urgent unmet medical need. Artificial intelligence (AI) has been demonstrating remarkable success in medical image analysis owing to the rapid progress of “deep learning (DL)” algorithms [11], which have shown increasing ability in solving complex and real-world problems in computer vision and image analysis. The possibility of digitizing gigapixel whole-slide images (WSIs) of tissues has led to AI and machine learning tools in digital pathology, which enable mining of subvisual morphometric phenotypes and may ultimately improve the patient therapeutic effect [12]. The combination of AI and the WSIs from tissue microarrays (TMAs) enables high throughput screening of a large number of patients. In this study, we have built a new DL-based precision oncology frameworkfrom immunostained TMA WSIs to accurately predict bevacizumab therapeutic effect in patients with EOC and PSPC. Importantly, the results show that therapeutic prediction is achievable without guidance or the use of prior knowledge of EOC or PSPC pathology in training AI models. Instead of directing the focus toward traditional pathological evaluation for tumor subclassification (e.g., papillary or clear cells formation, presence of mucin) or immunostaining features (e.g., percentage, intensity, score), the AI learning process is simply guided with the patient therapeutic effect data. In evaluation, the proposed model in combination with AIM2 achieves high accuracy 0.92, recall 0.97, F-measure 0.93 and AUC 0.97 for the first experiment (66% training and 34%testing) and high accuracy 0.86 ± 0.07, precision 0.9 ± 0.07, recall 0.85 ± 0.06, F-measure 0.87 ± 0.06 and AUC 0.91 ± 0.05 for the second experiment using five-fold cross validation, respectively. Both Kaplan–Meier PFS analysis and Cox proportional hazards model analysis further confirm that the proposed AIM2-DL model is able to distinguish patients gaining positive therapeutic effects with low cancer recurrence from patients with disease progression after treatment (p<0.005).

## 2. Related Works

### 2.1. Selection of Antibodies

Rather than directly targeting cancer cells, bevacizumab targets the tumor microenvironment, the effects of VEGF-inhibition are likely tumor-type and microenvironment-specific including the modulation of cancer immunity [13]. Cancer cells affect their microenvironment by releasing extracellular signals, thereby inducing tumor angiogenesis, and improving immune tolerance, thereby avoiding being recognized by the immune system. VEGF signaling supports immune suppression, and targeting VEGF/VEGFR has been recognized as an approach to enhance antitumor immunity in cancer patients [13]. Chronic inflammation perpetuated by inflammasome activation may play a central role in immunosuppression, angiogenesis, tumor proliferation, and metastasis. Conversely, inflammasome signaling also contributes to tumor suppression, which indicates the diverse roles of inflammasomes in tumorigenesis [14]. Inflammasomes are activated upon cellular infection that triggers the maturation of proinflammatory cytokines to engage innate immune defenses [15]. Once innate immune system related NOD-like receptors (NLRs) or AIM2-like receptors (ALRs) are activated, inflammation via the recruitment of immune cells such as macrophages promotes the proteolytic cleavage and secretion of proinflammatory cytokines (IL-1β and IL-18) through the activation of caspase-1, leading to cell senescence, apoptosis and the prevention of cancer progression [16,17,18]. In a previous study, the high expression level of AIM2 and NLRP3 was significantly correlated with poor PFS and disease progression of EOC which demonstrated a key role of the dysregulated inflammasome in modulating the malignant transformation of endometriosis-associated ovarian cancer [19]. In ascites of EOC patients, local complement activation has been observed to induce high complement anaphylatoxins level [20]. The immune genes involved in the complement system have dual influences on the survival of patients. Immunohistochemical analysis showed that the expression of the C3a receptor (C3aR) and the C5a receptor (C5aR) is higher in ovarian clear cell carcinoma [21]. Complement-activated factors have been related, either directly or indirectly, to neovascularization in several diseases [22]. The antiangiogenic factor was upregulated in monocytes by complement activation [23]. In contrast, a role for complement in the activation of angiogenesis has been demonstrated in age-related macular degeneration [24]. As the resistance to anti-VEGF treatment is related to immunity, in this study we explore the utility of four antibodies, including AIM2, NLPR3, C3 and C5, to differentiate patients with good treatment responses from patients with disease progression on EOC and PSPC.

### 2.2. Deep Learning in Application to Gynecologic Oncology

With an increase in computing power and advances in imaging technologies, DL is being implemented for the diagnosis and classification of medical images. Wang et al. [25] proposed a DL-based noninvasive recurrence prediction model in high-grade serous ovarian cancer (HGSOC) that extracts prognostic biomarkers from preoperative computed tomography (CT) images. Sato et al. [26] successfully applied DL to the classification of images from colposcopy. Matsuo et al. [27] compared the performance of DL models in survival analysis for women with newly diagnosed cervical cancer with conventional Cox proportional hazard regression (CPH) models. Ke et al. [28] proposed a DL diagnostic system that can distinguish high grade squamous intraepithelial lesion (HSIL), squamous cell carcinoma, atypical squamous cells of undetermined significance (ASCUS) and low grade squamous intraepithelial lesion. Wu et al. [29] introduced automatic classification of ovarian cancer types from cytological images using deep convolutional neural networks. Ghoniem et al. [30] built a multimodal evolutionary DL model for ovarian cancer diagnosis. Hong et al. [31] built multiresolution deep learning models for predicting endometrial cancer subtypes and molecular features from histopathology images. These studies demonstrate that gynecologists are able to utilize DL in clinical practice, increasingly.

### 2.3. Weakly Supervised Learning

The development of decision support systems for medical applications with deployment in clinical practice has been hindered by the need for large manually annotated datasets. To overcome problems of limited amount of data supervision, recent studies have investigated weakly supervised learning technologies. Campanella et al. [32] built a weakly supervised multiple instance deep learning system that uses only the reported diagnoses as labels for training accurate classification models in pathology and avoids expensive and time-consuming pixelwise manual annotations. Li et al. [33] presented an ensemble learning scheme to derive a safe prediction by integrating multiple weakly supervised learners to deal with inaccurate supervision, such as label noise learning, where the given labels are not always ground-truth. Kim et al. [34] develoep a weakly-supervised DL algorithm that diagnoses breast cancer at ultrasound without image annotation. Liu et al. [35] evaluated a weakly supervised deep learning approach to breast magnetic resonance imaging (MRI) assessment and showed that it is feasible to assess breast MRI images without the need for pixel-by-pixel segmentation using the weakly supervised learning method to yield a high degree of specificity in lesion classification. Lu et al. [36] built a weakly supervised clustering-constrained-attention multiple-instance learning (CLAM) model for data-efficient WSI processing and learning that only requires slide-level labels. These studies demonstrate that weakly supervised learning assists development and deployment of decision support systems for medical applications.

## 3. Materials and Methods

### 3.1. Study Population and Experimental Setup

A total of 720 sample tissue cores of 12 TMAs with four immune-related proteins, including inflammasome absent in melanoma (AIM2), nucleotide-binding domain leucine-rich repeat and pyrin domain containing receptor 3 (NLRP3), the complement of C3 and C5, were constructed with clinical information collected from March 2013 to January 2021 from the tissue bank of the department of pathology, Tri-Service General Hospital, National Defense Medical Center, Taipei, Taiwan,. Ethical approvals were obtained from the research ethics committee of the Tri-Service General Hospital (TSGHIRB No.1-107-05-171 and No.B202005070). The medical data were de-identified and used for a retrospective study without impacting patient care.

Patients in this study were divided into the bevacizumab-resistant or the bevacizumab sensitive group. Patients with persistently high levels of CA-125 during bevacizumab therapy or who experienced tumor progression or recurrence (assessed by CT/PET imaging) within six months posttreatment were classified as the bevacizumab-resistant group. Patients with low levels of CA-125 and no tumor progression or recurrence (based on imaging) during or within six months of bevacizumab treatment were classified as the bevacizumab sensitive group.

In data preparation, tissues from bevacizumab-treated EOC and PSPC patients were embedded in Paraffin wax. Two pathologists screened histological sections and selected areas of representative tumor cells, and one tissue core (2 mm in diameter) was then taken from each of the representative tumor samples and placed in a new recipient paraffin block for immunohistochemistry staining. The TMA sections were dewaxed in xylene, rehydrated in alcohol, and immersed in 3% hydrogen peroxide for 10 min to suppress the activity of endogenous peroxidase. Antigen retrieval was carried out by heating each section to 100 °C for 30 min in 0.01 M sodium citrate buffer (pH 6.0). The sections were rinsed three times (5 min each wash) in phosphate-buffered saline (PBS) and then incubated for one hour at room temperature with antibodies of AIM2 (1:500) (Abcam, cat#ab93015, Cambridge, UK), NLRP3 (1:300) (Millipore, cat#ABF23, Burlington, VT, USA), C3 (1:1000) (Abcam, cat#ab200999, Cambridge, UK) and C5 (1:300) (Abcam, cat#ab217027, UK) in PBS. The sections were washed three times (5 min each wash) in PBS, followed by incubation with horseradish peroxidase-labeled immunoglobulin (Dako, Carpinteria, CA, USA) for 1 h at room temperature. The sections were washed three times again, and the peroxidase activity was visualized using a solution of diaminobenzidine (DAB) at room temperature. Slides were stained with AIM2, NLRP3, C3 and C5 antibody. The WSIs were then acquired with a digital slide scanner (Leica AT Turbo) with a 20× objective lens.

In total, four datasets were built with four immune-related proteins. With regard to the class distribution, well-balanced data sets were ensured, and 57.2% of tissue cores were associated with effective Bevacizumab treatment outcomes, whereas 42.8% of tissue cores were associated with invalid treatment outcomes. The characteristics of the data are shown in Table 1. In evaluation, two experiments were conducted. For the first experiment, each dataset was split into two separate subsets for training and testing; 66% for training and 34% for testing, which ensured models were never trained and tested on the same sample (see Table 2). For the second experiment, a five fold cross validation was performed. In evaluation, models were evaluated independently on the testing set. In training, the batch size was set as six cores per batch with the learning rate as 0.003, and AI models were built independently for each stains. In evaluation, all models were independently evaluated on the test set.

### 3.2. Weakly Supervised Learning Framework

One of the obstacles for developing a deep neural network for medical image analysis application are data insufficiency. Small data may lead to under training, reducing the performance of the network, but large data cost enormous efforts in data collection and labeling. One of the contributions of this study is to develop an effective and efficient precision oncology system for prediction of therapeutic effect on ovarian cancer patients while utilizing as few manual annotations as possible. In this study, only the patient-based labels w.r.t. therapeutic effect on ovarian cancer patients and annotations on eight tissue cores, which account for less than 1% of the training image data, were utilized for building the system. We developed an efficient learning framework to produce AI models based on limited data with boosting learning, soft focusing sampling, boosted data augmentation and transfer learning. We built a single and robust weakly supervised model in selecting tumorlike tissues for all four kinds of immunohistochemical (IHC) stained data, which appear quite differently due to large variations on tissue morphology, staining magnitude, individual responses to antibodies and highlighting structures as shown in the high magnification views of Figure 1. The proposed weakly supervised tumorlike tissue selection model was built as follows.

#### Boosting Learning, Focusing Sampling, Boosted Data Augmentation and Transfer Learning

Transfer learning has been successfully applied to eliminate the problem of data scarcity in different biomedical image analysis applications. In transfer learning, a network captures knowledge from one problem and applies it to a different problem that contains a relatively small number of data samples for training the network properly. For producing the backbone network, only five H&E stained WSIs from IEEE automatic cancer detection and classification in whole slide lung histopathology challenge [37] were used. We hypothesized that this pretrain network may have the knowledge to identify the tissue morphology of tumors. Hence, this pretrained model was used as the backbone architecture for transferring knowledge to locate ovarian tumor tissues, and the model was retrained with our IHC images of the eight annotated tissue cores, which accounted for less than 1% of the training image data. However, to train a model with such a small dataset, a boosting learning approach was devised as follows.

Given a training set *S*:$\left\{\left(x_{i},y_{i}\right)\right\}$ where $x_{i}$ represents the instance data, and $y_{i}\in Y$:{0,1} represents the label, a learner ζ and the number of base models to build *U*, the proposed boosting learning produces the final AI model $\phi^{*}\left(x\right)$ by the following steps. First, create a new set $S_{1}$:$\left\{\left(t_{j},w_{j}^{1}\right)\right\}$ with instance weight $w_{j}$ where $t_{j}$:${\left\{x_{k}\right\}}_{k=1\cdots M\times M}$ represents a tile; M=512. Each instance weight $w_{j}^{1}$ is initialized with an IoU based attention weighting function ϖ for further training.
(1)$$w_{j}^{1}=\left\{\begin{array}{cc} 1, & \varpi \geq \alpha \\ 0, & otherwise \end{array}\right.$$
where $\varpi =\frac{\sum_{x_{i}\in t_{j}}y_{i}}{\mathrm{card}(t_{j})}$; α=0.05.

Then, iteratively for u=1…U, build a base model $\phi_{u}=\zeta \left(S_{u}\right)$. The sample weights $\left\{w_{j}^{u+1}\right\}$ are continuously modified and formulated by increasing the attention weights of false positives and false negatives of $\phi_{u}$.
(2)$$w_{j}^{u+1}=\left\{\begin{array}{cc} w_{j}^{u}+\chi , & \frac{\sum_{x_{k}\in t_{j}}{1|}_{\phi_{u}\left(x_{k}\right)\neq y_{k}}}{M\times M}\geq \alpha \\ w_{j}^{u}, & otherwise \end{array}\right.$$

Next, we devised a boosted data augmentation based on the sample attention weights $\left\{w_{j}^{u+1}\right\}$ and produced new data $S_{u+1}$. Data augmentation was applied to enlarge the training set with additional synthetically modified databy manipulating the rotation per 5° and 5 times and increment of 90°, the mirror-flipped along the horizontal and vertical axes, the contrast adjusted (random contrast, range 0% ± 20%), the saturation adjusted(random saturation, range 0% ± 20%), and the brightness adjusted(random brightness, range 0% ± 12.5%).

When the training data is partially labeled, many unlabeled tissues of interest are wrongly defined as background or contents of no interest. This severely confuses AI learners during supervised learning and deteriorates the performance output of AI models. To deal with this issue, we added an IoU-based focusing sampling mechanism for computing the gradients effectively. A number of unlabeled cells will now not be used as negative samples for training to confuse learning but arranged as ignored samples. This not only helps learning be more focused but also speeds up learning time. Moreover, we increased learning efforts on false positive and false negative predictions and further added variations of the FPs and FNs to assist AI learn better, deal with its weakness and produce improved AI models. An illustration of the proposed weakly supervised learning framework is given in Figure 2b.

For the learner ζ, the segmentation network is an extended version of our previous efforts in fast screening of cervical cancer [38] and thyroid cancer diagnosis [39]. The modified model is improved from its previous edition after including focusing sampling, boosted data augmentation and SoftMax, which is added to improve the prediction score for each class.

### 3.3. A Hybrid Deep Learning Model

In this study, a hybrid DL precision oncology framework was built, consisting of three DL models and two mapping models to rapidly process a gigapixel WSI of a TMA in seconds. Figure 2a presents the system workflow. First, a TMA core detector located tissue cores on a single patch in low resolution level (Section 3.3.1). Second, a forward-mapping function was applied to fetch the high resolution core data to be processed by a robust weakly supervised tumo-like tissue selection model (Section 3.3.2). Third, a backward-mapping function was applied to fetch the medium resolution tumorlike data of each core to be processed by a treatment effectiveness classification model (Section 3.3.3). The network architectures of the proposed hybrid DL framework are shown in Figure 3.

#### 3.3.1. Tissue Core Detection Model

The tissue core detection model rapidly locates tissue cores in low resolution. In our preliminary study, we compared the performance of cascade region-based convolutional neural networks (Cascade R-CNN) [41] and Faster R-CNN [42] for object detection, and it was found that Faster R-CNN often fails detecting objects that are close to each other, while Cascade R-CNN performs well in detection of those objects. Therefore, the TMA cores detection model was built with Cascade R-CNN. We formulated the two dimensional input data I into a multiresolution pyramid data structure with multiple levels *M* from high to low magnification ${\left\{I_{m}\right\}}_{m=0}^{M}$. The detector model $\Theta_{cores}$ aims to rapidly locate TMA cores on the low-magnification level $I_{\xi }$, generating a set of cores L using Equation (Equation 3).
(3)$$\begin{array}{c} L=\Theta_{cores}\left(I_{\xi }\right)={\left\{b_{\xi }^{d}\right\}}_{d=1}^{N} \end{array}$$
where *N* denotes the number of detected cores, and $b_{\xi }^{d}=\left(x_{\xi }^{d},y_{\xi }^{d},w_{\xi }^{d},h_{\xi }^{d}\right)$ represents the bounding box of a detected cores in level ξ.

Next, forward mapping was conducted to acquire each core data $b_{0}^{d}$ in the high magnification level $I_{0}$ for further tumor segmentation, generating a set of cores $H={\left\{b_{0}^{d}\right\}}_{d=1}^{N}\in I_{0}$.
(4)$$b_{0}^{d}=\left(x_{0}^{d},y_{0}^{d},w_{0}^{d},h_{0}^{d}\right)=2^{\xi }\left(x_{\xi }^{d},y_{\xi }^{d},w_{\xi }^{d},h_{\xi }^{d}\right)$$
where $b_{0}^{d}$ represents the bounding box of a detected cores in level 0 after forward mapping.

The tissue core detection model $\Theta_{cores}$ is a multistage object detection DL architecture [41] that consists of one region proposal network (RPN) and three Faster R-CNN [42] detectors. RPN proposes a set of low quality candidate bounding boxes from an image to determine the occurrence of an object, and therefore the subsequent detectors were developed to be more selective for lower quality candidates. The detector at stage ϑ consists of a bounding box-regressor $\Psi_{\vartheta }$ and a classifier $\Upsilon_{\vartheta }$. The bounding box-regressor $\Psi_{\vartheta }\left(a^{d},b^{d}\right)$ aims to regress a candidate bounding box $b^{d}$ of an object proposal image $a^{d}$ into a referenced bounding box $g^{d}$ and is learned from a training set $\left(b^{d},g^{d}\right)$ by minimizing the risk formulated as follows:(5)$$R_{loc}\left(\Psi_{\vartheta }\right)=\sum_{d}L_{loc}\left(\Psi_{\vartheta }\left(a^{d},b^{d}\right),g^{d}\right)$$

The TMA cores detection model was developed based on a cascaded bounding box-regressor Ψ formulated as follows:(6)$$\Psi \left(a,b\right)=\Psi_{\eta }\circ \Psi_{\eta -1}\circ \ldots \circ \Psi_{1}\left(a,b\right)$$
where η indicates the total number of stages. For multitask learning, each detector at stage ϑ is learned with the loss formulated as follows:(7)$$L\left(a_{\vartheta },g\right)=L_{cls}\left(\Upsilon_{\vartheta }\left(a_{\vartheta }\right),c_{\vartheta }\right)+\lambda \left[c_{\vartheta }\geq 1\right]L_{loc}\left(\Psi_{\vartheta }\left(a_{\vartheta },b_{\vartheta }\right),g\right)$$
where $b_{\vartheta }=\Psi_{\vartheta -1}\left(a_{\vartheta -1},b_{\vartheta -1}\right)$, λ is the trade-off coefficient, $c_{\vartheta }$ the label of $a_{\vartheta }$ under the intersection-over-union (IoU) threshold $\varphi_{\vartheta }$ with $\varphi_{\vartheta }>\varphi_{\vartheta -1}$ and [·] the Iversion bracket indicator function. In this study, three detectors η=3 with the IoU threshold φ={0.5,0.6,0.7} and λ=1.

#### 3.3.2. Weakly Supervised Tumorlike Tissue Selection Model

The tissue selection model aims to sample information of critical tissues for further analysis; information about how the weakly supervised tumorlike tissue selection model is built is described in Section 3.2. The tumorlike tissue selection model $\Theta_{tumor}$ was performed on each core in the patch-based data structure, producing tumor selection results for each tile $t_{j}^{\prime }\in b_{0}^{d}$.
(8)$${\left(p\left(x,y\right)\right)}^{c}=\Theta_{tumor}\left(t_{j}^{\prime }\right)$$
where c={0,…,C} denotes the number of types of tissue to be identified, and 0, 1 and 2 represent the background, others and tumorlike tissue, respectively.

A two dimensional pixel-based class map for each tile was produced as the index of the tissue type that has the maximum probability of the pixel using Equation (Equation 9).
(9)$$\kappa \left(x,y\right)=\underset{c}{\arg\max}\left({\left(p\left(x,y\right)\right)}^{c}\right)$$

Next, the non-tumorlike tissues were suppressed using Equation (Equation 10), and tiles with the qualified number of tumorlike tissues were selected for further training or testing the treatment effectiveness prediction model described in the subsequent section using Equation (Equation 11).
(10)$$\begin{array}{cc} t_{j}^{*}\left(x,y\right) & =\left\{\begin{array}{cc} t_{j}^{\prime }\left(x,y\right), & \kappa (x,y)>1\\ Ø, & otherwise \end{array}\right. \end{array}$$
(11)$$\begin{array}{cc} \pi_{j}\left(x,y\right) & =\left\{\begin{array}{cc} t_{j}^{*}\left(x,y\right), & \frac{\sum_{x,y}\kappa \left(x,y\right)}{\mathrm{card}(x,y)}>\alpha \\ Ø, & otherwise \end{array}\right. \end{array}$$

#### 3.3.3. Treatment Effectiveness Prediction Model

The treatment effectiveness prediction model of the proposed hybrid DL framework utilizes information of selected tumorlike tissues instead of nonmalignant cells such as stroma or background, and generates treatment effectiveness prediction (Figure 2(vii)). The treatment effectiveness prediction model was built based on Inception V3 [43], and modifications were made in the Inception Module 2 and 3. The network architecture is illustrated in Figure 3c where the blue highlighted convolution layers are the modified layers, for which the bias filter and scale filter are removed to avoid data distortion. Obtaining the selected tiles and tissues $\left\{\pi_{j}\right\}$ at level 0, backward mapping was performed to fetch the tumor features $\left\{\pi_{j}^{*}\right\}$ of each core from the middle-magnification level *l*, which is defined in Equation (Equation 12).
(12)$$l=\left\lfloor \frac{\log_{2}\left(w_{0}\times h_{0}\times 10^{-2\rho }\right)}{2}\right\rfloor$$
where ρ was set to 3 in the study.

The treatment effective decision model $\Theta_{classifier}$ applied onto the probability $\beta_{d}$ using the Equation (Equation 13). The task is to predict the treatment effectiveness outcomes $D=\{D_{1},\ldots ,D_{N}\}$ for individual cores, where *d* is the number of the cores; $D_{d}\in D=\left\{Invalid,Effective\right\}$ is the prediction of treatment effectiveness on the *d*-th core using Equation (Equation 14). In this study, δ was set to 0.5.
(13)$$\beta_{d}=\Theta_{classifier}\left(\pi_{j}^{*}\right)$$
(14)$$D_{d}=\left\{\begin{array}{cc} Effective, & \beta_{d}\geq \delta \\ Invalid, & \beta_{d}<\delta \end{array}\right.$$

#### 3.3.4. Model Selection Method with Early Stop Mechanism

A model selection method was developed in this study, and an illustration of the proposed model selection approach is given in Figure 4. Through training iteration *k*, the proposed model selection method computed the loss values $\omega_{k}$ and F-measure scores $\iota_{k}$ of trained models $M_{k}$ on the training set and further calculated associated first and second derivatives, including $\left(\omega_{k}^{\prime },\omega_{k}^{\prime \prime }\right)$ and $\left(\iota_{k}^{\prime },\iota_{k}^{\prime \prime }\right)$.
(15)$$\omega_{k}^{\prime }=\frac{\partial \omega_{k}}{\partial k}$$
(16)$$\omega_{k}^{\prime \prime }=\frac{\partial^{2}\omega_{k}}{\partial k^{2}}$$
(17)$$\iota_{k}^{\prime }=\frac{\partial \iota_{k}}{\partial k}$$
(18)$$\iota_{k}^{\prime \prime }=\frac{\partial^{2}\iota_{k}}{\partial k^{2}}$$

Next, $\left(\omega_{k}^{\prime },\omega_{k}^{\prime \prime }\right)$ are continuously evaluated for every δ iterations to find models with stable loss where $\left(\omega_{k}^{\prime },\omega_{k}^{\prime \prime }\right)$ converges to [−ϵ,ϵ] for a continuous period, the starting time $k^{s}$ of the stable loss window is used as the starting point in the model selection search window as shown in Figure 4a,b. Afterwards, the model searches for a stable F-measure where $\left(\iota_{k}^{\prime },\iota_{k}^{\prime \prime }\right)$ converges to [−τ,τ] for a continuous period, an early stop mechanism is activated to stop training and set the end of the search window of model selection as $k^{e}$; otherwise, $k^{e}$ is set as the maximum number of training times given as an input parameter. Then, a model $M_{i^{*}}$ is selected in the model selection search window by finding a model with the highest F-measure score on the training set. If there are multiple models with the highest score, the one with the largest training iteration time is selected. (δ,ϵ,τ are input parameters and empirically defined as 1000, 0.003 and 0.1, respectively).
(19)$$\left\{i\right\}=\arg\max_{k^{s}\leq j\leq k^{e}}\iota_{j}$$
(20)$$i^{*}=\arg\max_{i}\left\{i\right\}$$

## 4. Results

### 4.1. Quantitative Evaluation

For the first experiment (see Table 3), the Proposed-AIM2 model achieved the highest recall 0.97, AUC 0.97, F-measure 0.93 and accuracy 0.92, respectively, and Coudray-AIM2 obtained the highest precision 0.97 and AUC 0.97, respectively. It was also found that Coudray-AIM2 and Coudray-C3 obtained good performance with F-measure equal to 0.91. Figure 5a further compares the ROC curves on the testing set for all models, showing that the Proposed-AIM2 model consistently performed well, regardless of the value of the selected threshold. Moreover, Figure 5b compares the performance of AUC through iterations of AI models without model selection on the testing set, showing that the Proposed-AIM2 model generally outperformed other models with the same training time.

For the second experiment using five-fold cross validation (see Table 4), the proposed-AIM2 model was demonstrated to achieve the highest accuracy 0.86 ± 0.07, precision 0.9 ± 0.07, recall 0.85 ± 0.06, F-measure 0.87 ± 0.06 and AUC 0.91 ± 0.05. Overall, the proposed method obtained better results using AIM2 than C3, C5 and NLRP3. The results of the two experiments indicate that AIM2 could be an effective biomarker for guiding ovarian cancer treatment.

### 4.2. Statistical Analysis

Studies of how patients respond to treatment over time are critical to investigating how therapies influence disease progression during survivorship, and generally two closely related statistical analyses, i.e., Kaplan–Meier (K-M) analysis and Cox proportional hazards model analysis, are performed [45] where K-M is a univariate approach, while Cox analysis is a multivariable approach. In this study, the results of the K-M analysis and Cox proportional hazards model analysis are provided as follows.

#### 4.2.1. Kaplan–Meier Progression Free Survival and Overall Survival Analysis

To further examine the performance of the proposed precision oncology models, we conducted Kaplan–Meier analysis in PFS and overall survival (OS) time using the log-rank test. The binary prediction outcomes (0: invalid, 1: effective) of each model were used to categorize patients into two groups, and then a Kaplan–Meier curve was generated to visualize the probability of a patient having disease progression as time increases. Figure 6a compares the Kaplan–Meier PFS curves of the top two models in Table 3, showing the proposed-AIM2 as highly effective (p=0.004) and Coudray-AIM2 as effective(p=0.038) to distinguish patients gaining positive therapeutic effects with low cancer recurrence from patients with disease progression after treatment with statistical significance. On the other hand, Figure 6b compares the Kaplan–Meier OS curves of the two top models, showing that both models are not effective to distinguish treatment effectiveness with respect to the overall survival time.

#### 4.2.2. Cox Proportional Hazards Model Analysis

We further investigated DL model prediction and clinical factors, including age, BMI, number of treatments, cancer stage (FIGO), histology and surgery type, in association with disease progression using multivariate analysis. However, as shown in Table 5, only the proposed DL model prediction is useful as an indicator for patient selection with statistical significance (p<0.01); patients who were predicted to be effective by the proposed-AIM2 DL model had a lower risk of recurrence than patients predicted to be invalid (HR=0.18,p=0.003).

## 5. Conclusions and Discussion

This study presents a new precision oncology approach with a hybrid DL framework to guide cancer treatment on patients with EOC and PSPC using immunostained TMA WSIs. Bevacizumab is the first antiangiogenic agent to have demonstrated benefit as first-line and maintenance therapy in EOC and was explored in two large prospective randomized trials. The Gynecologic Oncology Group 218 and ICON 7 phase III trials revealed significantly prolonged PFS for carboplatin/paclitaxel plus bevacizumab followed by bevacizumab maintenance versus carboplatin/paclitaxel alone [46]. Other antiangiogenics have been assessed in EOC, such as pazopanib, sorafenib, sunitinib, cediranib, VEGF trap (aflibercept), and AMG386. However, EOC response rates to these inhibitors were relatively low, ranging from 6–20% [47]. In addition, none have been adopted in routine clinical practical use due to toxicity and the cost of licensing [9,48]. Bevacizumab is the only approved antiangiogenic agent in routine clinical practice.

Unlike most other targeted therapies, bevacizumab is used in the general patient populations instead of targeting patients preselected by a biomarker. Despite intense efforts, no predictive biomarker has been identified for personalized use of bevacizumab [49]. Given the high cost, the potential for toxicity, and finding that only a subset of patients will benefit from these drugs, patients who receive bevacizumab treatment should be carefully selected [50]. Although several potential predictive biomarkers such as circulating combination of Ang1 and Tie2 proteins, VEGFR-1 and neuropilin-1 in the plasma or cancer tissue as well as plasma cell-free DNA [51,52,53], and Ang-2, FGF, HGF activation of c-Met, deltalike ligand 4 (Dll4) -induced Notch signaling, hedgehog signaling or inhibition of Zeste homolog 2 (EZH2) [54,55,56,57], are found, these biomarkers have not consistently been predictive of response.

In the past decade, advances in precision oncology have resulted in an increased demand for predictive methods that enable the selection of patients for treatment. In clinical practice, anatomical pathologists base their histological diagnosis on the visual recognition, semiquantification and integration of multiple morphological features of the analyzed sample. However, the traditional histology-based classification of EOC has no benefit for the prediction of therapeutic outcome. Immunohistochemistry (IHC) with antibodies specific for EOC antigens help identify those that are overexpressed or aberrantly expressed in tumor tissues and may be a potential biomarker for therapeutic prediction. In our previous study [58], the expression of AIM2, C3 and C5 was analyzed using immunohistochemical staining according to the percentage and intensity of the color reaction by visual observation to predict the efficacy of bevacizumab in epithelial ovarian cancer (EOC) patients, and the AIM2 immunostaining scores were significantly higher in the bevacizumab-resistant group than in the bevacizumab-sensitive group (*p* < 0.001), but there were no significant differences in C3 (*p* = 0.077) or C5 (*p* = 0.326) regarding bevacizumab. Based on the results of the previous study and this study, we propose a possible link between the AIM2 inflammasome and antiangiogenic therapy in EOC. In the microenvironment favorable for EOC, the inflammasome promotes transcription of certain NLRs. NLRs and AIM2 assemble into the inflammasome complex which via the caspase recruitment domain, recruit platelets and circulating leukocytes, all of which can secrete proangiogenic factors and promote angiogenesis [19].

In cancer, the complexity of genomic alterations that affect cell signaling and cellular interactions with their environment can influence the biological course of the disease and affect responses to therapeutic interventions. The assessment of such alterations requires simultaneous interrogation of multiple features with highly sensitive and precise approaches [59]. The development of new AI-based image analysis approaches in pathology and oncology is being led by computer engineers and data scientists, who are developing and applying AI tools for a variety of tasks such as helping improve diagnostic accuracy and identify novel biomarker approaches for precision oncology [12]. Over the past few years, interest in the use of machine learning-based approaches for drug discovery and development has increased [60]. The fact that most patients who receive cytotoxic agents or immune checkpoint inhibitors do not respond to treatment has led to increasing interest in combining AI with digital pathology to identify the patients who are most likely to derive therapeutic benefit [12,60]. The previous report has also developed an approach whereby the spatial arrangement of nuclei or tumor-infiltrating lymphocytes enables prediction of the responsiveness of patients with late-stage non-small-cell lung cancer to the antiprogrammed cell death 1 (PD-1) antibody nivolumab [12]. Machine learning can not only be used to analyze the tumor cells of epithelial origin, increasing interest exists in trying to identify prognostic patterns within the tumor environmental stroma cells [61].

To the authors’ best knowledge, this is the first precision oncology framework created to effectively predict the bevacizumab therapeutic effect of EOC or PSPC patients. The proposed-AIM2 model is demonstrated to achieve excellent performance on accuracy 0.92, recall 0.97, F-measure 0.93 and AUC 0.97. Furthermore, results of both Kaplan–Meier PFS analysis and Cox proportional hazards model analysis show that the proposed model can distinguish patients gaining positive therapeutic effects with low cancer recurrence from patients with disease progression after treatment (p<0.01). This study indicates new potential precision oncology systems to predict antiangiogenic therapy benefits in EOC and PSPC patients for assisting personalized medicine are possible.

Furthermore, the proposed method could also be applied to other types of cancers for angiogenesis inhibitors, such as metastatic breast cancer, non-small-cell lung cancer, glioblastoma, renal cell carcinoma, and cervical cancers. We hope that DL could play a role in immunohistochemical stains for other potential candidate proteins such as Ang1, Ang-2, Tie2, VEGFR-1, neuropilin-1, FGF and EZH2 to develop more effective predictive biomarkers for personalized antiangiogenic target therapies. In conclusion, we demonstrate that DL could be a very useful tool for assisting AIM2 immunostained slides in applying the appropriate and tailored targeted therapy, increasing the scope and performance of precision medicine, which aims to develop a treatment method tailored to the patients with EOC and PSPC. Therefore, AI approaches will become key in analyzing these large amounts of data, helping pathologists and oncologists in the process. These results highlight the emerging role of DL in precision medicine for therapeutic prediction and suggest an expanding utility for computational analysis of histology in the future practice of pathology.

## Figures and Tables

**Figure 1 cancers-14-01651-f001:**
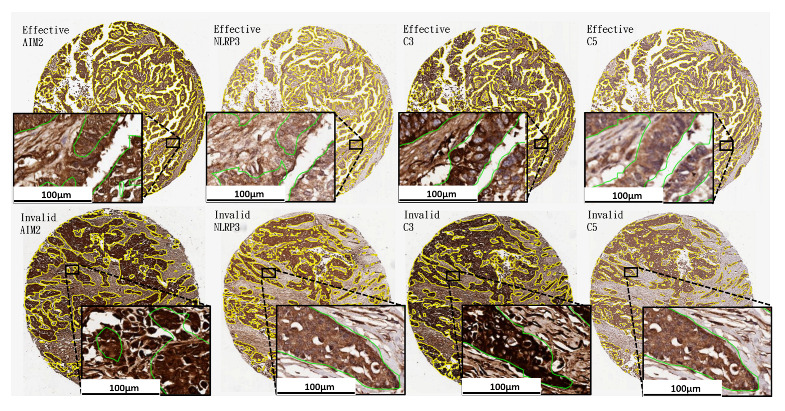
Sample tumor-like tissue selection results with high magnification views at 100 μm of the four kinds of IHC stained data by the proposed weakly supervised tissue selection model.

**Figure 2 cancers-14-01651-f002:**
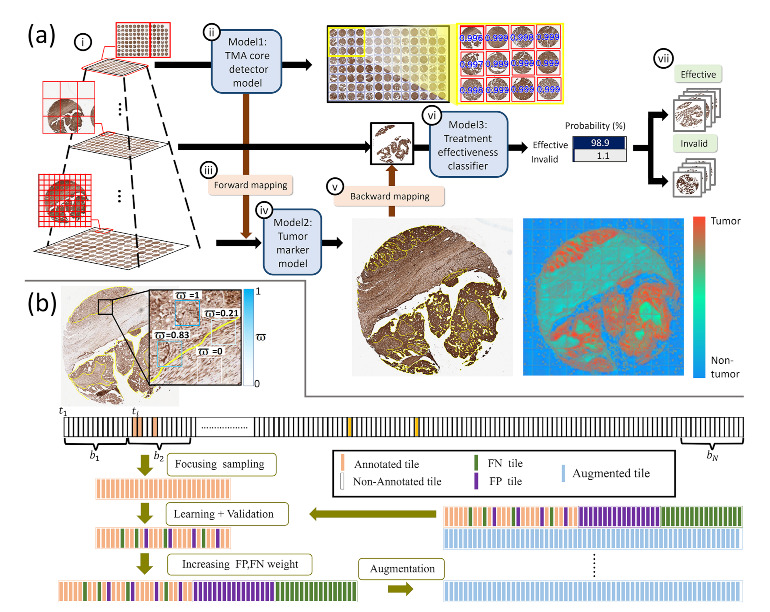
(**a**) System workflow: (i) multiresolution pyramid data structure of WSIs; (ii) a TMA core detection model conducts fast localization of tissue cores in low-resolution level; (iii) a forward-mapping function is applied to fetch the high resolution core data to be processed by (iv) a robust tumorlike tissue selection model to locate tumorlike tissues of each core; (v) a backward-mapping function is applied to fetch the medium resolution tumorlike tissue data of each core to be processed by; (vi) a treatment effectiveness classification model to predict; (vii) treatment outcomes; (**b**) Weakly supervised learning with focusing sampling, boosting learning and boosted data augmentation.

**Figure 3 cancers-14-01651-f003:**
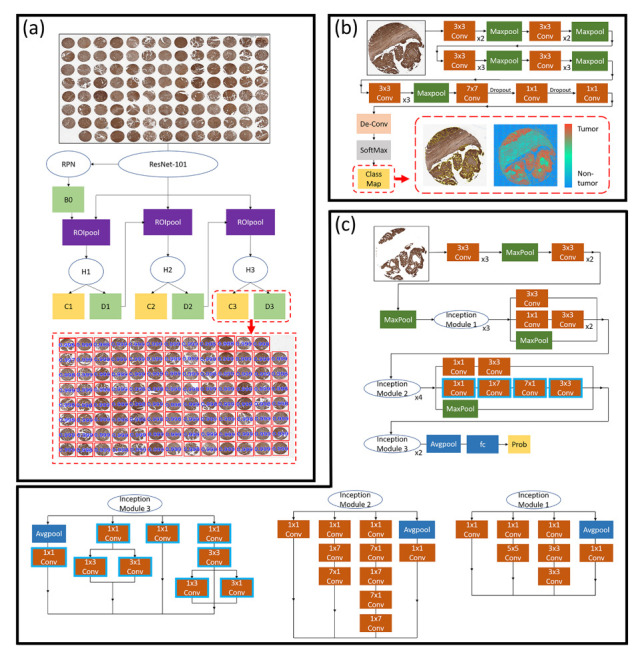
The proposed hybrid deep learning precision oncology framework contains three deep learning networks. (**a**) In the tissue core detection model, ResNet-101 backbone where RPN proposes a set of low quality candidate bounding boxes (B0) from an image to determine the occurrence of an object, and therefore the subsequent detectors are developed to be more selective for lower quality candidates. The modules (H) produced samples for training the z-th classifier and detector. Ref. [40] is used as the RPN (**b**) The proposed weakly supervised tumorlike tissue selection model composed of 16 convolution layers (each convolution layer is followed by a RELU layer), five pooling layers for downsampling, and an upsampling layer. The class map of the core is generated by a deconvolution layer and a SoftMax layer. (**c**) The proposed treatment effectiveness prediction model uses dimensional reduction and parallel structures of the inception modules and contains three different sizes of convolution and one maximum pooling. For the network output of the previous layer, the channel is aggregated after the convolution operation, and then the nonlinear fusion is performed.

**Figure 4 cancers-14-01651-f004:**
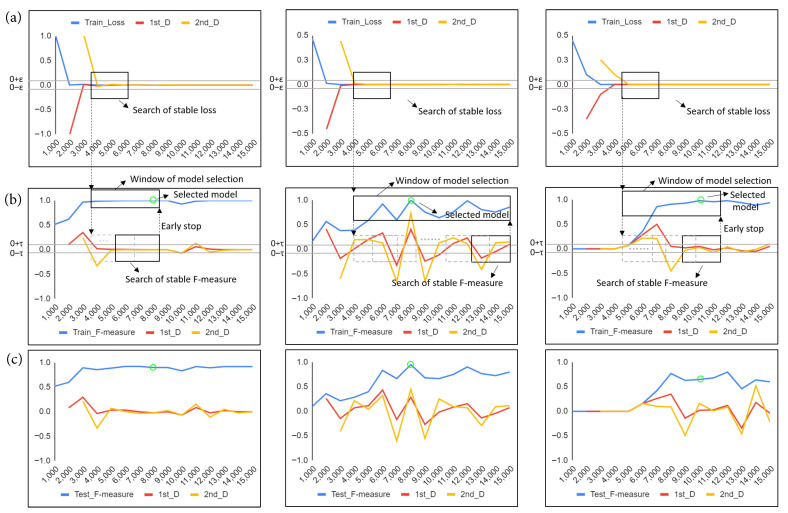
Illustration of the proposed model selection method with three examples: (**a**) loss values of models through iterations during training; (**b**) F-measure scores of individual models with different iterations on the training set; (**c**) F-measure scores of individual models with different iterations on the testing set. The blue lines indicate the measurement values, and the red and yellow lines represent the associated first and second derivatives, respectively. During training, the proposed model selection method computes the loss values and F-measure scores of trained models on the training set and further calculates associated first and second derivatives. If a stable loss is found in (**a**) where the first and second derivatives of the training loss converge for a continuous period, the starting point of the stable loss window is used as the starting point in model selection searching. Afterwards, if a stable F-measure is found in (**b**) where the associated first and second derivatives converge for a continuous period, an early stop mechanism is activated to stop training and end the search window of model selection. Then, a model is selected in the model selection search window by finding a model with the highest F-measure score on the training set. If there are multiple models with the highest score, the one with the largest training iteration time is selected. Green circles highlight model selection results using the F-measure on the training set in (**b**) and demonstrates that overall the selected models obtain relatively high F-measure scores on the testing set in (**c**).

**Figure 5 cancers-14-01651-f005:**
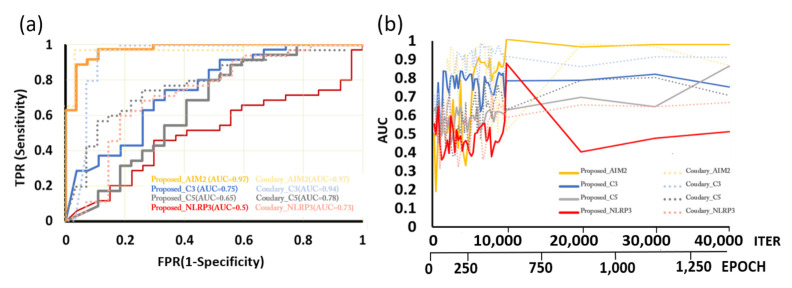
(**a**) Receiver operating characteristic (ROC) curves on the testing set for the models of the proposed method and the benchmark approach [44]; (**b**) Graphs of AUC on the testing set with respect to the iteration times in training among the DL models, showing that the Proposed-AIM2 model generally outperforms other models with the same training time.

**Figure 6 cancers-14-01651-f006:**
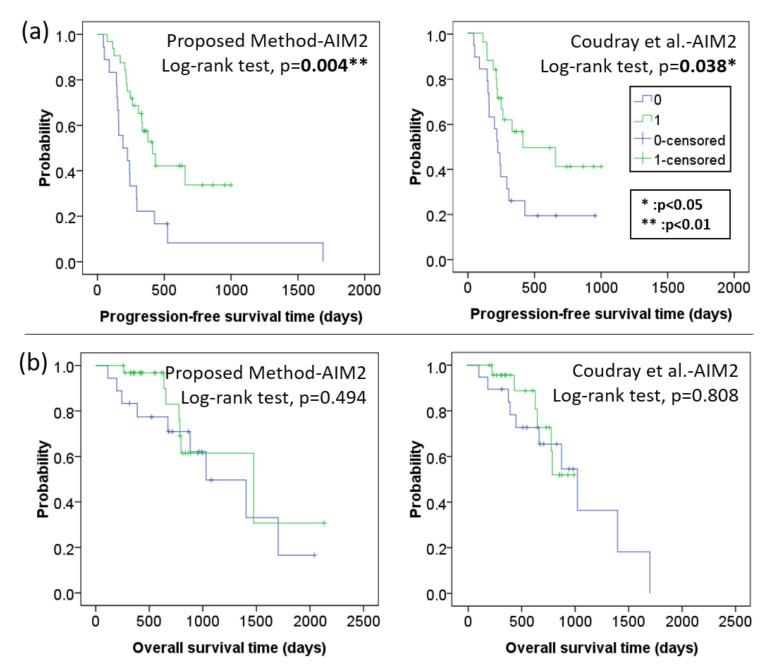
(**a**) Kaplan–Meier PFS and; (**b**) OS analysis for EOC and PSPC patients receiving bevacizumab therapy based on AI prediction outcomes (0: invalid; 1: effective) by the two best models, i.e., proposed method-AIM2 and Coudray et al. [44]-AIM2.

**Table 1 cancers-14-01651-t001:** Baseline characteristics of data.

Characteristics	*N*
Tissue Core	720
Patient age (mean, range)	(59.1, 23–79)
Diagnosis (%)	
Papillary serous carcinoma	444 (61.6)
Peritoneal serous papillary carcinoma	89 (12.3)
Clear cell carcinoma	69 (9.6)
Unclassified carcinoma	69 (9.6)
Endometrioid carcinoma	39 (5.5)
MC	10 (1.4)
FIGO stage (%)	
I	69 (9.6)
II	39 (5.4)
III	454 (63)
IV	158 (22)
Surgery (%)	
Optimal debulking	306 (42.5)
CRS+HIPEC	217 (30.1)
Suboptimal debulking	197 (27.4)
Treatment effectiveness (%)	
Effective	412 (57.2)
Invalid	308 (42.8)

**Table 2 cancers-14-01651-t002:** Data distribution of the collected four datasets w.r.t. immune-related proteins for training and testing for the first experiment.

	Treatment Outcome	AIM2	C3	C5	NLRP3
Training (66%)	Effective	68	68	68	68
	Invalid	50	50	50	50
Testing (34%)	Effective	35	35	35	35
	Invalid	27	27	27	27
Total		180	180	180	180

**Table 3 cancers-14-01651-t003:** First experiment:Quantitative evaluation in classification of therapeutic outcomes.

Method in Combination with Potential Biomarker	Accuracy	Precision	Recall	F-Measure	AUC
Proposed Weakly Supervised DL Method—AIM2	**0.92**	0.89	**0.97**	**0.93**	**0.97**
Coudray et al. [44]—AIM2	0.90	**0.97**	0.86	0.91	**0.97**
Proposed Weakly Supervised DL Method—C3	0.69	0.69	0.83	0.75	0.75
Coudray et al. [44]—C3	0.90	0.91	0.91	0.91	0.94
Proposed Weakly Supervised DL Method—C5	0.63	0.67	0.69	0.68	0.65
Coudray et al. [44]—C5	0.69	0.68	0.86	0.76	0.78
Proposed Weakly Supervised DL Method—NLRP3	0.52	0.56	0.71	0.63	0.50
Coudray et al. [44]—NLRP3	0.71	0.76	0.71	0.74	0.73

**Table 4 cancers-14-01651-t004:** Second experiment: 5-fold cross-validation.

Method in Combination with Potential Biomarker	Accuracy	Precision	Recall	F-Measure	AUC
ProposedWeakly Supervised DL Method—AIM2	**0.86 ± 0.07**	**0.9± 0.07**	**0.85 ± 0.06**	**0.87 ± 0.06**	**0.91 ± 0.05**
Coudray et al. [44]—AIM2	0.73 ± 0.17	0.76 ± 0.19	0.71 ± 0.39	0.68 ± 0.33	0.9 ± 0.07
Proposed Weakly Supervised DL Method—C3	0.75 ± 0.1	0.77 ± 0.1	0.79 ± 0.1	0.78 ± 0.09	0.78 ± 0.12
Coudray et al. [44]—C3	0.73 ± 0.08	0.77 ± 0.06	0.74 ± 0.21	0.74 ± 0.1	0.84 ± 0.09
Proposed Weakly Supervised DL Method—C5	0.65 ± 0.03	0.66 ± 0.03	0.8 ± 0.04	0.72 ± 0.02	0.66 ± 0.07
Coudray et al. [44]—C5	0.56 ± 0.13	0.69 ± 0.22	0.51 ± 0.3	0.54 ± 0.22	0.52 ± 0.23
Proposed Weakly Supervised DL Method—NLRP3	0.56 ± 0.08	0.59 ± 0.05	0.77 ± 0.08	0.67 ± 0.06	0.55 ± 0.08
Coudray et al. [44]—NLRP3	0.63 ± 0.18	0.68 ± 0.19	0.66 ± 0.32	0.63 ± 0.28	0.73 ± 0.24

**Table 5 cancers-14-01651-t005:** Multivariate analyses of DL model prediction and clinical factors associated with recurrence.

	Adjusted HR ${}^{1}$ (95% C.I.)	*p* Value
Age	1.03 (0.98–1.07)	0.212
BMI	1.01 (0.92–1.11)	0.848
Number of bevacizumab used times	0.97 (0.89–1.05)	0.472
FIGO ${}^{2}$ (III + IV vs. I + II)	4.23 (0.82–21.86)	0.085
Histology (others vs. serous)	0.82 (0.26–2.62)	0.737
Surgery		
CRS + HIPEC ${}^{3}$	1.00 (reference)	reference
optimal	0.92 (0.35–2.42)	0.869
suboptimal	1.18 (0.42–3.30)	0.753
Theraphy		
Concurrent therapy	1.00 (reference)	reference
Second-line therapy	1.71 (0.66–4.41)	0.265
Maintenance therapy	0.23 (0.04–1.40)	0.110
DL model prediction		
Proposed DL Method-AIM2 (effective v.s. invalid)	0.18 (0.06–0.55)	**0.003 ***

^1^ HR = Hazard ratio; ^2^ FIGO = International Federation of Gynecology and Obstetrics. ^3^ CRS+HIPEC = Cytoreductive surgery with hyperthermic intraperitoneal chemotherapy. * The proposed model prediction is useful as an indicator for patient selection with statistical significance (*p* < 0.01).

## Data Availability

The data that support the findings of this study are available from the corresponding author upon reasonable request.

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
