# Peer review of "A Weakly Supervised Deep Learning Method for Guiding Ovarian Cancer Treatment and Identifying an Effective Biomarker"

_cancers, 2022, doi:10.3390/cancers14071651_

Round 1
Reviewer 1 Report
In this manuscript, Wang et. al. introduced a weakly supervised DL model that could screen and select EOC and PSPC patients with potential positive molecular therapies. Specifically, the model built with AIM2 IHC sample data collected from the authors’ database can achieve an exceptionally high performance based on various statistical metrics. The authors also presented solid works from building up the cohort to DL architecture designing. The manuscript is overall well-written and the results are plausible. However, there are several questions and concerns to be addressed by the authors.
- While using transfer learning is a reasonable option, have the authors even tried training the model from scratch? It is possible that it can work better than transfer learning models, so the authors may want to show the comparison.
- In Fig4B, the figure panel and legends are confusing. On the left side it says “AUC” but the legend says loss.
- If it is AUC by iteration, NLPR3 model seems to have spike around 10000 iterations. Did authors consider early stop at that point and see how the model performs?
- If it is loss by iteration, seems like all models are fairly overfitted in an early stage. While this is typical for these WSI trained DL tasks, did the authors think about simplify their architecture or apply some regularizations to prevent overfitting?
- Regardless of true plot context, it would be better to show iteration equivalent epochs rather than iteration alone.
- The authors compared their model with the Coudray et. al. model extensively, which is necessary and leads to some questions:
- According to Table 2, among the 4 biomarkers selected, only on AIM2 did the authors’ method achieve a similar performance as Coudray et. al. I’d suggest the authors to dig further into why this happened and lower their tones on introducing their models.
- Coudray et. al. model also achieved very promising results on C3, which is comparable to AIM2. This should also be noted as a finding of the paper.
- NLRP3 works on Coudray et. al. model, but not the authors’. Why is that?
- Did the authors use OS or DSS in addition to PFS in accessing survival?
- Fig5B is misleading as Coudray et. al. model also achieved significant p-value in log-rank test. “*” and bold font should be put on it too.
- The introduction section still lack of recent publication literature reviews, which need to be modified. There are some other very interesting publications on introducing novel DL models to apply to oncology especially gynecological cancers, which the authors may have to add. One example would be “Predicting endometrial cancer subtypes and molecular features from histopathology images using multi-resolution deep learning models https://doi.org/10.1016/j.xcrm.2021.100400”
Overall, the manuscript can be accepted for publication if the authors address all these questions above.
Reviewer 2 Report
The authors proposed weakly supervised deep learning models for ovarian cancer treatment with help of effectively predicting biomarkers such as AIM2, c3, C5, and NLRP3 for patients diagnosed with epithelial ovarian cancer (EOC) and peritoneal serous papillary carcinoma (PSPC) ovarian cancer. However, the author needs to focus on some of the points identified as follows.
1. The article systematically presented the research work with a clear system workflow that fascinates the reader. Also, the boosted data augmentation and statistical analysis further improve the standard of the article.
2. As the author proposed a weakly supervised deep learning model for identifying effective biomarkers for ovarian cancer, they should have many pieces of literature survey on it in section 2 of Related work. However, there is no literature on the weakly supervised model of machine learning or deep learning.
3. The cohort doesn’t provide any indication about the demographic information of the age group. The treatment and biomarker may differ for the different age groups.
4. The author does not consider the cross-validation method to validate their results. K-5 or K-10 cross-validation is the standard cross-validation protocol. Its usage and comparison with the current results are required.
5. A “Hybrid Deep Learning Model” as used in the article should be theoretically defined briefly and cited before proceeding.
6. Need a few lines explanation regarding the results using performance parameters “Precision” of the proposed model versus other compared studies.
Round 2
Reviewer 1 Report
The authors have made substantial efforts in addressing my previous comments and concerns. I would therefore recommend for publication in present form. Thanks!